# Analysis of the Influence Mechanism of Consumers' Trading Behavior on Reusable Mobile Phones

**Qingbin Yuan** [1,2], **Yifan Gu** [1,2], **Yufeng Wu** [1,2,*], **Xinnan Zhao** [1,2] **and Yu Gong** [1]

1    College of Materials Science and Engineering, Beijing University of Technology, Beijing 100124, China;
     yuanqingbin@emails.bjut.edu.cn (Q.Y.); guyifan@bjut.edu.cn (Y.G.); zxn@crra.com.cn (X.Z.);
     gongyu@bjut.edu.cn (Y.G.)
2    Institute of Circular Economy, Beijing University of Technology, Beijing 100124, China
*    Correspondence: wuyufeng3r@126.com; Tel./Fax: +86-10-67392345

**Abstract:** The aim of this study is to investigate the decision-making mechanism of reusable mobile phone trading behaviors by using the extended theory of planned behavior. In this study, based on the survey data of 964 residents in Beijing, China and structural equation modeling method, the main factors that affect consumers' reusable mobile phone trading behavior and their degree of influence were analyzed, followed by discussion on decision-making mechanisms. The findings show that consumers' behavioral selection has been significantly related to four intrinsic subjective factors and fifteen external objective factors, and the combined effect of the latter ones is nearly triple of that of the former ones. Moreover, the observed variables of environmental awareness, information leakage sensitivity, trading convenience and consumer trading returns are the four most significant factors. The impact of active trading behavior is not significant. However, this may be because that there were no great trading rewards, lack of trading awareness and regulations. Finally, the study put forward relevant policy recommendations for improving the comprehensive management of recycling reusable mobile phones, and provides a theoretical reference for improving the recycling rate of reusable mobile phones.

**Keywords:** reusable mobile phone; extended theory of planned behavior; trading behavior; mechanism of influence; recycling rate

## 1. Introduction

The development of the Internet has accelerated the frequency of mobile phone updates. For example, the frequency of replacement of mobile phones by half of Chinese users has reached about 18 months [1], which has greatly increased the number of reusable mobile phones. Only in 2018, the number of reusable mobile phones in China reached 830 million, nearly three times higher than in 2010 [2]. The mobile phone scraps put new demands on the recycling capacity of urban electronic waste. To this end, the Chinese government has successively promulgated the "Regulations on the Management of the Recycling of Waste Electrical and Electronic Products", the "Subsidy Standards for Trading of Waste Electrical and Electronic Products", the "Administrative Measures on the Collection and Use of Waste Electrical and Electronic Products Processing Funds" and other laws and regulations. The trading behavior is guided by policy [3,4]. However, such measures have not fundamentally changed the difficulty of recycling reusable mobile phones in China. For example, by 2015, the recycling rate of reusable mobile phones in China was still lower than 2% [5]. Subsequently, the state added subsidies for the e-waste dismantling process, and successively issued special fund policies such as the "Promoters of the Extended Producer Responsibility System", but the results seemed unsatisfactory [6,7]. Until March 2016, mobile phones were classified into the latest Waste

Electrical and Electronic Equipment (WEEE) management catalog. Up to now, China has not announced clear subsidy specifications and policies, resulting in slow progress in the trading process of reusable mobile phones. In addition to the government's intervention in trading of reusable mobile phones, the success of the recycling project depends on the support of trading times of consumers and recycling facilities [8,9]. In previous literature, most researchers explore the trading mode of consumers as key subjects or major stakeholders in the recycling system of reusable mobile phones [10,11]. However, fully encouraging consumers to trade is a necessary step for establishing a complete recycling system. Few published studies have been able to draw on any systematic research into reusable mobile phone trading behaviors which includes all possible reusable mobile phone trading methods simultaneously. In order to improve the recycling rate of reusable mobile phones, this paper discusses the factors affecting consumers' trading behavior of reusable mobile phones, explores consumers' behavioral preferences, and analyzes the relevance of various influential factors to the trading behavior of reusable mobile phones in combination with the extended plan behavior theory. At the same time, it proposes to improve the comprehensive management-related policies for trading reusable mobile phones.

The results of this study could provide theoretical support for policy formulation on the recycling rate of reusable mobile phones in China and other countries in the world. The initial research hypotheses and conceptual models are organized in Section 2, based on previous literature reviews. Section 3 introduces the research methodology for structural equation modeling. Section 4 presents model testing and results. The conclusion, policy recommendations, future research and limitations are conducted in Section 5.

## 2. Theoretical Model Construction

In recent years, domestic and foreign scholars have conducted related research [12,13]. Through a literature review, the paper summarizes the factors that have a greater impact on consumers. Some scholars summarized the trading model of consumers as the main body of recycling systems, others explored the influence of multiagent and multi-influence factors on the recycling system of consumers [14–16]. Based on the theoretical model of extended plan behavior, this paper constructs the decision-making factor model of reusable mobile phone trading behavior, including behavioral attitude, subjective norm, perceptual behavior control, and four aspects of environmental facility services [8,17]. The model of the structural equation analysis of consumer reusable mobile phone trading behavior (RMTB) is constructed by combining the subjective elements of the extended plan behavior theory (Figure 1).

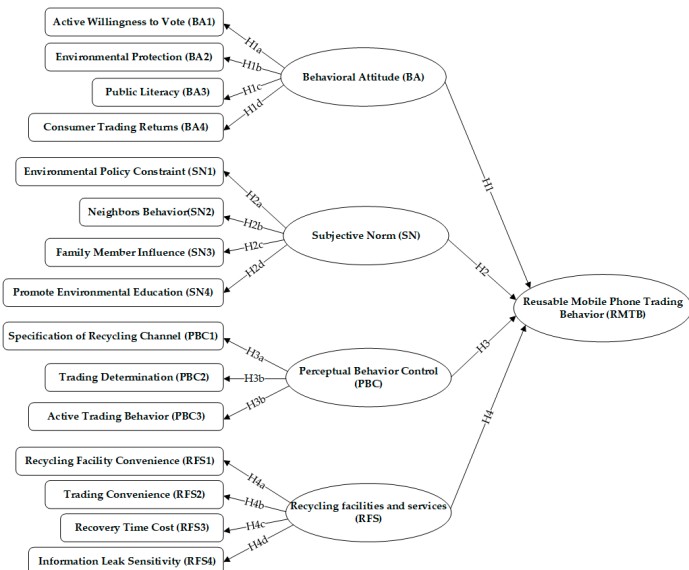

**Figure 1.** Reusable mobile phone processing behavior model.

In the model, the study assumes four behavioral variables: behavioral attitude (BA), subjective norm (SN), perceptual behavior control (PBC), and recycling facility and service (RFS). Trading behaviors (H1, H2, H3, and H4) have path effects, and each observed variable is reflected by several observed variables. This paper establishes 15 possible observed variables based on literature surveys.

## 2.1. Behavioral Attitude (BA)

Behavioral attitude refers to the positive or negative attitude of an individual to a certain behavior, and the individual's attitude towards this particular behavior is evaluated through conceptualization [18,19]. According to other scholars' research, there is a significant correlation between behavioral attitudes and consumers' trading behavior of reusable mobile phones [20–22]. The attitude to the trading of used mobile phones will be affected by consumer factors [23]. In addition, research shows that if the government does not provide trading incentives to consumers, consumers will not actively reduce the waste generated, indicating that economic factors will also affect consumers' trading behavior [24].

In this study, the following four factors are used to explain the influence of behavioral attitudes on their trading behavior: Active Willingness to Vote (BA1), Environmental Protection (BA2), Public Literacy (BA3) and Consumer Trading Returns (BA4). The four factors can reflect its impact on behavioral attitudes to a large degree. Therefore, hypothesis 1: the attitude of consumers to mobile phones has a significant positive impact on the trading behavior of reusable mobile phones.

## 2.2. Subjective Norm (SN)

Subjective norm refers to the social pressures an individual perceives for a particular behavior. In other words, individuals are prepared to perform specific behaviors, regardless of whether they perform specific behaviors that are influenced by individuals or groups that influence individual behavior decisions [4,16]. It is considered that subjective norm has a significant correlation with consumers' trading behavior [25]. Specifically, important influential factors focus on following the psychological role, social identity, civil society responsibility, and the constraints of laws and regulations [26,27]. Research shows that environmental knowledge and information mastery have a significant positive correlation with consumers' implementation of trading behavior, which means that strengthening public education and intensity will have a strong positive correlation with trading participation rate [11].

The following four factors are used to explain the impact of subjective norms on consumer behavior: Environmental Policy Constraint (SN1), Neighbors' Behavior (SN2), Family Member Influence (SN3), and Promote Environmental Education (SN4). Therefore, hypothesis 2: the subjective norm of consumers' mobile phones has a significant positive impact on the trading behavior of reusable mobile phones.

## 2.3. Perceptual Behavior Control (PBC)

Perceptual behavior control refers to the individual perception of past experience and future expectation of a particular behavior. When individuals feel that they have a lot of resources and opportunities, and they are expected to take a little hindrance to a behavior, the control of perceived behavior will increase and they will be more likely to take action [28,29]. Chen used Beijing residents as an example to analyze the factors affecting the willingness to trade [30]. The research results show that the promotion of environmental protection and the improvement of relevant laws and regulations promotes consumers' active participation in recycling [31]. Ramayah not only believes that trading habits have a significant positive impact on the recycling intention of discarded electronic products in urban counties in China, but also verifies the impact of the realization of recycling, and raises the awareness of residents on the trading of reusable mobile phones [32]. It is suggested that the higher the residents' awareness of trading, the stronger their willingness to participate in trading,

and the increased awareness of trading can promote the reasonable recycling of electronic waste to some extent [33].

The following three factors are used to explain the impact of perceptual behavior control on consumer behavior: Specification of Recycling Channel (PBC1), Trading Determination (PBC2) and Active Trading Behavior (PBC3). Therefore, hypothesis 3: consumers' perceived behavioral control cognition of mobile phones has a significant positive impact on the trading behavior of reusable mobile phones.

### 2.4. Recycling Facilities and Services (RFS)

The increase in renewable resource recycling sites has helped to increase recovery rates [34,35]. Ouyang suggested that the anxiety related to consumer transaction security will have a negative impact on the use of products in the mobile payment service of smartphones [36,37]. This shows that the instability of product safety factors causes consumer concerns, so consumers will give up use of the product [38]. External factors affecting the trading behavior of reusable mobile phones using recycling facilities and service variables are explained by the following four variables: Recycling Facility Convenience (RFS1), Trading Convenience (RFS2), Recovery Time Cost (RFS3), Information Leak Sensitivity (RFS4). So, hypothesis 4 can be formulated thus: personal information leakage and the time cost of recycling have a significant negative impact on the reusable mobile phone trading behavior.

## 3. Research Methodology

### 3.1. Data Sources

Based on the theoretical model (Figure 1), a questionnaire in Appendix A was designed using a 7-point Likert scale to investigate RMTB and its influencing factors. The questionnaire consists of three parts: (1) the demographic and social attribute information of the respondent, including gender, age, education level, family monthly income, and residential area; (2) the status quo of reusable mobile phone consumers according to different reusable mobile phone processing behavior, which is divided into three categories: shelving at home, trading, discarding; (3) measuring the initial research hypotheses and conceptual model of the project; all measures were reported on a 7-point scale. For the detailed classification, see Supporting Information.

Data was collected through questionnaires (Table 1). The survey was launched in the areas where residents gather, including eight shopping malls in downtown Beijing, JoyCity, TAIKOOLI and Wangfujing Street. It fully considered the population ratio in each district, as well as the characteristics of distribution of residents of all age groups and with different occupations. The questionnaire survey collected 964 valid questionnaires out of 1038 in total, with a valid response rate of 92.82%.

We chose Beijing as the research area because Beijing has the most comprehensive reverse supply chain system for reusable mobile phones in China, including reusable e-commerce platforms, recycling companies, recyclers, and dismantling companies. In addition, Beijing's economy is developed and mobile phone ownership is high. Whether it can solve the problem of e-waste is related to Beijing's image and reputation. Therefore, the Beijing Municipal Government and the public are more eager to comply with environmental protection, and are willing to cooperate with the investigation. The establishment and implementation of relevant laws and regulations are also easy to carry out.

**Table 1.** Measurement instruments for the latent variables of the hypothetical model.

| Variable Dimension | Serial Number | Questionnaire Topic Design | Resources |
|---|---|---|---|
| Behavioral attitude (BA) | BA1 | Adopting reusable mobile phones will help us enhance the quality of the environment at home. | [18] |
| | BA2 | In order to save resources and protect the environment, I am willing to participate in reusable mobile phone trading. | [19,24] |
| | BA3 | Trading reusable mobile phones is conducive to saving resources. | [23] |
| | BA4 | A paid treatment will encourage us to adopt it. | [39] |
| Subjective norm (SN) | SN1 | Reusable mobile phone recycling laws and regulations can play a constraining role for me. | [40] |
| | SN2 | What my family and neighbors think we should do is important to me. | [41] |
| | SN3 | I feel morally obliged to reduce the volume of untreated mobile phones discharged into the environment. | [25] |
| | SN4 | Our relatives support the idea of us adopting the reusable mobile phones. | [11] |
| Perceptual behavior control (PBC) | PBC1 | Trading reusable mobile phones actively if we are comfortable with the recycling technology. | [28] |
| | PBC2 | Collecting reusable mobile phones takes up a lot of storage space in my house. | [29] |
| | PBC3 | A simple and easy to operate procedure will encourage us to trade. | [42] |
| Recycling facilities and services (RFS) | RFS1 | A free mobile phone trading treatment will encourage us to trade it. | [38] |
| | RFS2 | There are trading bins in the community, with clear identification and within close distance. | [36] |
| | RFS3 | Low-time-cost treatment will encourage us to adopt it. | [43] |
| | RFS4 | I attach great importance to the disclosure of information during the transaction. | [44] |

### 3.2. Measurement Model

The structural equation model (SEM) used in this study and the software AMOS 22.0 derived the initial model of the concept. SEM was first proposed by the scholar Joreskog in the 1970s [45]. The structural equation model consists of a structural model and a measurement model. The structural model is used to estimate the relationship between latent variables [46]. The measurement model is used to determine the relationship between latent variables and observed variables. SEM combines the advantages of statistical methods such as factor analysis, path analysis, and multiple regression. The general form of the structural equation model is shown in Equation (1):

$$\begin{cases} X = \lambda_x \xi + \delta \\ Y = \lambda_y \eta + \varepsilon \end{cases} \tag{1}$$

where: $\xi$ is an exogenous latent variable; $\eta$ is an endogenous latent variable; $X$ is the observed variable matrix of $\xi$; $Y$ is the observed variable matrix of $\eta$; $\lambda_x$ is the relationship between the dependent variable and the exogenous latent variable; $\lambda_y$ is the self-relationship between the variable and the endogenous latent variable; $\delta$ is the error term of $X$; $\varepsilon$ is the error term of $Y$.

The final calculation model is shown in Equation (2):

$$\eta = \gamma\xi + \beta\eta + \zeta \tag{2}$$

where: $\gamma$ is the relationship between the exogenous latent variable and the endogenous latent variable; $\beta$ is the relationship between endogenous latent variables; $\zeta$ is the error term of the structural equation.

We use the software AMOS to evaluate total effects of each predictor variable on the endogenous variables and conduct the model calculation using maximum likelihood estimation (MLE). In short, the structural equation model in this study consists of three steps: (1) reliability and validity testing of survey data; (2) confirmatory factor analysis (CFA) to assess the effectiveness of the measured structure; (3) model evaluation.

## 4. Data Analysis and Results

### 4.1. Statistical Sample

The statistical analysis was a questionnaire survey with 964 valid responses. The characteristic distribution of the collected samples is shown in Table 2. Only part of the content is listed here.

**Table 2.** Distribution of the socio-demographic characteristics of the samples.

| Social Attribute Characteristics | | Total Number of Samples | |
|---|---|---|---|
| | | Frequency | Proportion (%) |
| Gender | Male | 439 | 45.51 |
| | Female | 526 | 54.49 |
| Age [1] | Under 18 | 49 | 5.12 |
| | 18–30 | 331 | 34.34 |
| | 31–60 | 565 | 58.56 |
| | 61 and above | 20 | 2.03 |
| Education level [1] | Junior high school and below | 116 | 11.98 |
| | High school, secondary school | 516 | 53.43 |
| | University specialties, undergraduate | 174 | 18.07 |
| | Graduate and above | 159 | 16.52 |
| Family monthly income [2] | Under 5000 | 225 | 23.31 |
| | 5000–8000 | 380 | 39.38 |
| | 8001–20,000 | 304 | 31.50 |
| | 20,000 and above | 56 | 5.81 |
| Number of phones in use | 0 | 6 | 0.58 |
| | 1 | 628 | 65.12 |
| | 2 | 296 | 30.72 |
| | 3 and above | 34 | 3.57 |

[1] Age and education level are obtained based on the data of the Sixth Population Census in 2010. [2] The average wage of employees in Beijing (2020) was 6906 Yuan.

The distribution of socio-demographic characteristics of valid samples is comparable to the total population distribution in Beijing, indicating that the survey has good credibility.

Based on the above analysis and adjustment results, this paper uses AMOS 22.0 to construct the initial model of the structural equation analysis of consumer reusable mobile phone trading behavior (RMTB) [47]. It uses the maximum likelihood estimation method to estimate the parameters of the model. The initial evaluation results are shown in Figure 2. The research hypotheses H1, H2, H3, and H4 are all confirmed. The process of consumers participating in mobile phone trading behavior intentions follow the path of "cognition → trading decision". This form is influenced by precognitive factors such as behavioral attitude (AB), subjective norm (SN), perceived behavioral control (PBC), and recycling facilities and services (RFS).

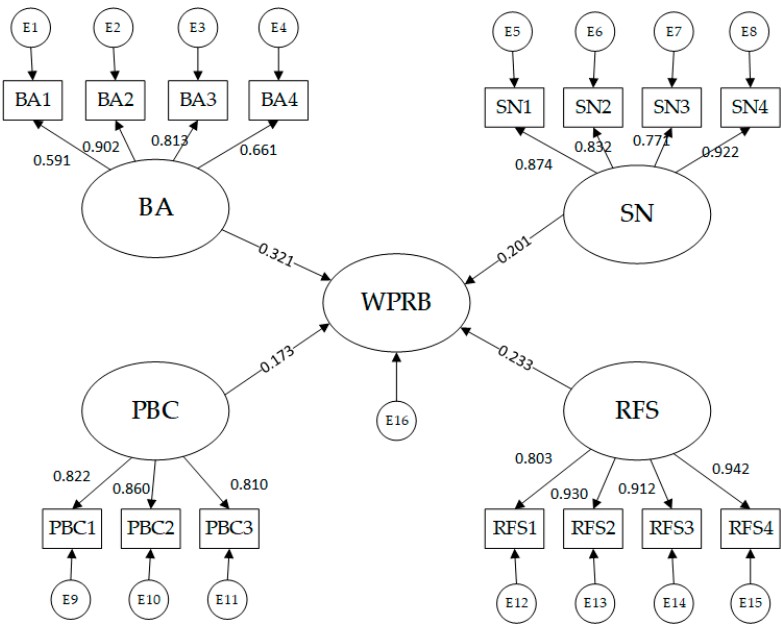

**Figure 2.** Evaluation results of parameter estimates for the structural equation model.

This reliability and validity of the study are evaluated. The composite reliability (CR) is used to test the convergent reliability of the measurement. As shown in Table 3, the CR for each latent variable is greater than 0.8. Because the composite reliability of all latent variables is more than 0.6, the measurement of this study is acceptable in reliability [48]. This study used average variation extracted (AVE) to test the convergent validity of the measurement. Average variation extracted (AVE) calculates how well observed questionnaire items of a construct explain the average variation of the construct. As shown in Table 3, the AVEs of the four latent variables are 0.702, 0.783, 0.736, and 0.766, respectively. Since the AVEs of the four latent variables are greater than 0.5, this indicates that the convergent validity of the measurement is acceptable [49].

**Table 3.** The indicators of reliability and validity of the measurement model.

| Variable Dimension | Serial Number | Mean | Standard Deviation | Factor Load($\lambda$) | CR [1] | AVE [2] |
|---|---|---|---|---|---|---|
| Behavioral attitude | BA1 | 5.632 | 0.865 | 0.591 * | 0.831 | 0.702 |
| | BA2 | 5.163 | 0.892 | 0.902 ** | | |
| | BA3 | 5.062 | 0.806 | 0.813 ** | | |
| | BA4 | 5.421 | 0.808 | 0.661 ** | | |
| Subjective norm | SN1 | 5.236 | 0.763 | 0.874 ** | 0.893 | 0.783 |
| | SN2 | 5.219 | 0.784 | 0.832 *** | | |
| | SN3 | 5.024 | 0.824 | 0.771 * | | |
| | SN4 | 5.096 | 0.830 | 0.922 ** | | |
| Perceptual behavior control | PBC1 | 5.087 | 0.756 | 0.822 ** | 0.862 | 0.736 |
| | PBC2 | 5.632 | 0.718 | 0.860 ** | | |
| | PBC3 | 4.962 | 0.861 | 0.810 *** | | |
| Recycling facilities and services | RFS1 | 4.896 | 0.721 | 0.803 * | 0.872 | 0.766 |
| | RFS2 | 5.621 | 0.722 | 0.930 * | | |
| | RFS3 | 5.032 | 0.767 | 0.912 ** | | |
| | RFS4 | 5.067 | 0.874 | 0.942 ** | | |

[1] CR (Composite Reliability) = $\frac{(\sum \lambda)^2}{(\sum \lambda)^2 + (\sum P)}$; [2] AVE (Average Variance Extracted) = $\frac{\sum \lambda^2}{\sum \lambda^2 + \sum p}$; ($\lambda$ is the factor loading; $p$ is the observable variables' error variance; * $p < 0.05$, ** $p < 0.01$, *** $p < 0.001$).

To meet the requirement for discriminant validity, the square root of a latent variable's AVE must be higher than the correlations between the latent variable and the other variables in the study [50]. In Table 4, the diagonal elements are the square root values of AVEs, and the other elements are the Pearson correlation coefficients among the constructs. For example, the square roots of the AVEs for the two constructs of perceptual behavior control and subjective norm are 0.854 and 0.883, which is more

than the correlation 0.689, found between them in Table 4. It shows that there is adequate discriminant validity between the two latent variables. The square roots of all latent variables' AVEs of this study are all greater than the correlations among all constructs in Table 4. Thus, the discriminant validity of the measurement is acceptable.

**Table 4.** Pearson correlation coefficients and square root values of AVEs.

| Latent Variable | BA | SN | PBC | RFS |
|---|---|---|---|---|
| Behavioral attitude (BA) | 0.836 ** | | | |
| Subjective norm (SN) | 0.752 ** | 0.883 ** | | |
| Perceptual behavior control (PBC) | 0.823 ** | 0.689 ** | 0.854 ** | |
| Recycling facilities and services (RFS) | 0.623 ** | 0.766 ** | 0.693 *** | 0.871 ** |

Note: ** $p < 0.01$, *** $p < 0.001$.

### 4.2. Model Result Analysis

The overall fit of the model is divided into absolute fit indices, incremental fit indices, and parsimony fit indices [51]. For absolute fit indices, the chi-square value/df (degrees of freedom) = 2.101, which is less than 3; the goodness-of-fit index (GFI) = 0.907, which is more than 0.9; the possible range of GFI values is 0–1, with higher values indicating better fit. The standardized root mean residual (SRMR) = 0.029 and the root mean square residual (RMR) = 0.038, which indicated satisfactory fit. The root means square error of approximation (RMSEA) = 0.052. Lower RMR and SRMR values represent better fit and higher values represent worse fits, which puts the RMR, SRMR, and RMSEA into a category of indices sometimes known as badness-of-fit measures in which high values are indicative of poor fit.

For incremental fit indices, the normed fit index (NFI) = 0.953, which is more than 0.9, and a model with perfect fit would produce an NFI of 1. One disadvantage is models that are more complex will necessarily have higher index values and artificially inflate the estimate of model fit. So, calculating the following incremental fit measures: The comparative-fit index (CFI) = 0.916 and the relative noncentrally index (RNI) = 0.921, which is more than 0.9, represents an acceptable model fit.

For parsimony fit indices, the adjusted goodness-of-fit index (AGFI), and parsimony normed fit index (PNFI), relatively high values represent relatively better fit. AGFI values are typically lower than GFI values in proportion to model complexity. PNFI can be used in the same way as the NFI. PNFI is the widely applied parsimony fit index.

According to the results shown in Table 5, the overall fit of the model in this study is acceptable. The following table is the commonly used judgment standard of the fit index.

**Table 5.** Model fit test results.

| Statistical Test Indicator Type | Fit Goodness Statistics | Appropriate Fit Goodness Statistics | Standard Value |
|---|---|---|---|
| Absolute fit indices | $\chi^2$/df | 2.101 | <3.00 |
| | GFI | 0.907 | >0.90 |
| | RMSEA | 0.052 | <0.08 |
| | RMR | 0.038 | <0.05 |
| | SRMR | 0.029 | <0.05 |
| Incremental fit indices | NFI | 0.953 | >0.90 |
| | CFI | 0.916 | >0.90 |
| | RNI | 0.921 | >0.90 |
| Parsimony fit indices | AGFI | 0.904 | >0.90 |
| | PNFI | 0.528 | >0.50 |

4.2.1. Findings on the Impact of Participant Age on Trading Behavior

The SPSS 22.0 software was used to test the differences in different sub-dimensions of the four types of potential variables: behavioral attitude (H1), subjective norm (H2), perceived behavior control (H3), recycling facility and service (H4). The indicators of age group descriptive statistics (Table 6) indicate that it was not possible to statistically determine a correlation for young consumers (under 30) in Beijing more so than when compared to the elders (above 31).

**Table 6.** The indicators of age group descriptive statistics.

| Latent Variable | | Number | $\bar{x} \pm s$ |
|---|---|---|---|
| Behavioral attitude (H1) | under 30 | 380 | $4.132 \pm 0.765$ |
| | above 31 | 585 | $4.663 \pm 0.792$ |
| Subjective norm (H2) | under 30 | 380 | $4.462 \pm 0.706$ |
| | above 31 | 585 | $4.021 \pm 0.802$ |
| Perceptual behavior control (H3) | under 30 | 380 | $4.336 \pm 0.763$ |
| | above 31 | 585 | $4.219 \pm 0.754$ |
| Recycling facilities and services (H4) | under 30 | 380 | $4.094 \pm 0.724$ |
| | above 31 | 585 | $4.026 \pm 0.710$ |

Note: $\bar{x}$ is mean of categorical samples; s is standard deviation.

Due to the differences in each group, a homogeneity test of variances was performed before proceeding to the post hoc Tukey tests. Since the effectiveness factor is $p = 0.122 > 0.05$ ($p$ is the observable variables' error variance), the variance is homogeneously distributed. Pairwise comparisons (post hoc Tukey tests) were carried out for the factors that are found to be significant according to the analysis results in Table 7.

**Table 7.** Variance analysis results for the impact of age.

| Latent Variable | F | df | Significance |
|---|---|---|---|
| Behavioral attitude (H1) | 1.933 | 963 | 0.925 |
| Subjective norm (H2) | 1.521 | 963 | 0.024 |
| Perceptual behavior control (H3) | 2.822 | 963 | 0.126 |
| Recycling facilities and services (H4) | 3.682 | 963 | 0.457 |

The post hoc Tukey tests (Table 8) show that the age variable makes a difference on the dimension of Subjective norm ($p = 0.032 < 0.05$). Results indicate that the young consumers (under 30) in Beijing will actively reduce the generated waste more compared to the elders (above 31).

**Table 8.** The test result of post hoc Tukey tests on the impact of age.

| Dependent Variable | (I)Category | (J)Category | Average Difference(I-J) | Significance |
|---|---|---|---|---|
| Subjective norm (H2) | under 30 | above 31 | 0.441 | 0.032 |

4.2.2. Findings on Impact of Consumers' Education Levels on Trading Behavior

According to the results, there is a statistically significant difference of the consumer education independent variable on the behavioral attitude ($p = 0.025 < 0.05$) sub-dimension compared to 0.05 level of significance. A variance uniformity test result shows that the variance is homogenous since the effectiveness factor for equality factor is $p = 0.603 > 0.05$. Pairwise comparisons are made for the factors that are found to be significant according to the analysis results in Table 9.

**Table 9.** Variance analysis results for the impact of education levels.

| Latent Variable | F | df | Significance |
|---|---|---|---|
| Behavioral attitude (H1) | 3.393 | 963 | 0.025 |
| Subjective norm (H2) | 2.525 | 963 | 0.924 |
| Perceptual behavior control (H3) | 5.822 | 963 | 0.126 |
| Recycling facilities and services (H4) | 3.663 | 963 | 0.457 |

The post hoc Tukey test (Table 10) shows that the age variable makes a difference on the dimension of behavioral attitude ($p = 0.018 < 0.05$). Results indicate that ordinary consumers (bachelor's degree or below) in Beijing will actively reduce the generated e-waste more compared to the intellectual (bachelor's degree and above).

**Table 10.** The test result of post hoc Tukey tests on the impact of education levels.

| Dependent Variable | (I)Category | (J)Category | Average Difference(I-J) | Significance |
|---|---|---|---|---|
| Behavioral attitude (H1) | bachelor's degree or below | bachelor's degree and above | 0.553 | 0.018 |

### 4.2.3. Findings on the Impact of Consumers' Gender on Trading Behavior

Considering the strong independence between male and female in the independent variable of gender, we used an independent t-test to analyze the difference between the two subsamples. The study found that in the sub-dimensions of transaction behaviors of consumers participating in the study, the average value for females is higher than that for males (Table 11).

**Table 11.** The indicators of gender group descriptive statistics.

| Latent Variable | | Number | Mean |
|---|---|---|---|
| Behavioral attitude (H1) | Male | 439 | 3.412 |
| | Female | 526 | 3.532 |
| Subjective norm (H2) | Male | 439 | 3.062 |
| | Female | 526 | 3.321 |
| Perceptual behavior control (H3) | Male | 439 | 3.036 |
| | Female | 526 | 3.119 |
| Recycling facilities and services (H4) | Male | 439 | 3.326 |
| | Female | 526 | 3.496 |

When the results of the analysis presented in Table 12 are examined, it is seen that the gender of the consumer who participated in the research does not have a significant impact on the trading behavior perception sub-dimensions, compared to the significance level of 0.05.

**Table 12.** T-test analysis results for the impact of gender.

| Latent Variable | t | df | Significance |
|---|---|---|---|
| Behavioral attitude (H1) | 0.933 | 963 | 0.725 |
| Subjective norm (H2) | 1.121 | 963 | 0.221 |
| Perceptual behavior control (H3) | 0.922 | 963 | 0.236 |
| Recycling facilities and services (H4) | 1.082 | 963 | 0.657 |

### 4.2.4. Findings on Impact of Consumers' Family Income on Trading Behavior

After checking the results shown in Table 13, Independent variables of consumers' family income have statistically significant differences in the sub-dimension of recycling facilities and services ($p = 0.046 < 0.05$), while the significance level is 0.05. A variance uniformity test result shows that the variance is homogenous since effectiveness factor for equality factor is $p = 0.573 > 0.05$. Pairwise

comparisons are made for the factors that are found to be significant according to the analysis results in Table 13.

**Table 13.** Variance analysis results for the impact of family income.

| Latent Variable | F | df | Significance |
|---|---|---|---|
| Behavioral attitude (H1) | 3.133 | 963 | 0.825 |
| Subjective norm (H2) | 3.121 | 963 | 0.220 |
| Perceptual behavior control (H3) | 2.892 | 963 | 0.526 |
| Recycling facilities and services (H4) | 3.981 | 963 | 0.046 |

In the post hoc Tukey tests (Table 14), it is seen that the family income independent variable makes a difference on the recycling facilities and services sub-dimension of democracy perception. It is observed that amongst consumers in Beijing, those who can get a high income (above 8001) are more sensitive to the convenience of recycling facilities in the process of reusing mobile phones than those who get a low income (below 8000).

**Table 14.** The test result of post hoc Tukey tests on the impact of family income.

| Dependent Variable | (I)Category | (J)Category | Average Difference(I-J) | Significance |
|---|---|---|---|---|
| Recycling facilities and services (H4) | above 8001 | under 8000 | 0.322 | 0.017 |

4.2.5. Findings on the Impact of Consumers' Professional Status on Trading Behavior

After checking the results shown in Table 15, Independent variables of consumer professional status have statistically significant differences in the sub-dimension of perceptual behavior control ($p = 0.026 < 0.05$), while the significance level is 0.05. A variance uniformity test result shows that the variance is homogenous since the effectiveness factor for equality factor is $p = 0.573 > 0.05$. Pairwise comparisons are made for the factors that are found to be significant according to the analysis results in Table 15.

**Table 15.** Variance analysis results for the impact of professional status.

| Latent Variable | F | df | Significance |
|---|---|---|---|
| Behavioral attitude (H1) | 2.973 | 963 | 0.605 |
| Subjective norm (H2) | 4.121 | 963 | 0.584 |
| Perceptual behavior control (H3) | 3.122 | 963 | 0.026 |
| Recycling facilities and services (H4) | 1.782 | 963 | 0.357 |

In the post hoc Tukey tests (Table 16), it is seen that the professional status independent variable makes a difference on the perceptual behavior control sub-dimension of democracy perception. It is observed that amongst consumer in Beijing, those who work in the environmental protection industry are more sensitive to having the awareness of trading reusable mobile phones than those who work in the non-environmental protection industry.

**Table 16.** The test result of post hoc Tukey test on the impact of professional status.

| Dependent Variable | (I)Category | (J)Category | Average Difference(I-J) | Significance |
|---|---|---|---|---|
| Perceptual behavior control (H3) | environmental protection industry | non-environmental protection Industry | 0.441 | 0.038 |

4.2.6. Findings on Impact of Number of People Using Mobile Phones on Trading Behavior

According to the results, the independent variable of number of people using mobile phones constitutes a statistically significant difference on the recycling facilities and services ($p = 0.047 < 0.05$) sub-dimension compared to 0.05 level of significance. A homogeneity test shows that the effectiveness factor is $p = 0.065 > 0.05$ and variance is homogeneously distributed. Pairwise comparisons are made for the factors that are found to be significant according to the analysis results in Table 17.

**Table 17.** Variance analysis results for the impact of number of using mobile phones.

| Latent Variable | F | df | Significance |
|---|---|---|---|
| Behavioral attitude (H1) | 4.123 | 963 | 0.925 |
| Subjective norm (H2) | 2.518 | 963 | 0.054 |
| Perceptual behavior control (H3) | 3.222 | 963 | 0.126 |
| Recycling facilities and services (H4) | 2.182 | 963 | 0.047 |

The post hoc Tukey test shows that the independent variable of number of people using mobile phones does not cause a difference to the recycling facilities and services dimension of the organizational democracy perception. In terms of effectiveness, the pairwise comparisons could not statistically determine whether the number of people using mobile phones is more effective for the consumers regarding recycling facility convenience.

## 5. Discussion and Implications

### 5.1. Discussion of Results

In order to improve the recycling rate of reusable mobile phones, this study puts forward the SEM model of Extended Theory of Planned Behavior (ETPB) theory to construct the behavior mechanism of consumer reusable mobile phone trading.

This study makes important contributions to the literature of improving the recycling rate of reusable mobile phones by summarizing the main factors affecting consumers' trading behavior and the degree of impact. First of all, this study examines consumers' reusable mobile phone transaction behaviors from various occupations, such as the environmental protection industry, IT, and the financial industry, and this will reveal the practices of the mobile phone recycling industry. Improving consumer transaction returns and participation in reusable mobile phone transactions actively, in a sense, will also contribute to improving the recycling rate of reusable mobile phones. The study, which was conducted with both ordinary and high consumption consumers in Beijing, shows that there is a different sensitivity of behavior attitude, subjective norm, perceptual behavior control and recycling facilities and services, whereas the negative consciousness is not there yet. Previous studies pay more attention to the benefit gain of consumers that exceed the costs storing at home. Trading channel specification had a significant positive effect on the trading of reusable mobile phones [23,39]. Explaining the concept of reusable mobile phone trading behavior more from a theoretical background perspective, and the use of a new scale are the reasons why this study makes a contribution to this field.

The important findings obtained from the hypotheses of the study indicate that young consumers (under 30) in Beijing will actively reduce the generated e-waste more compared to the elders (above 31). Thus, they contribute to improving the recycling rate of reusable mobile phones in terms of reducing waste emissions. Similarly, the study shows that that amongst consumers in Beijing, those who work in the environmental protection industry are more sensitive to having the awareness of trading reusable mobile phones than those who work in the non-environmental protection industry. Compared to consumers with bachelor's degrees, more of those with Masters/PhD degrees state that they actively reduce the generated e-waste. This coincides with issues, such as consumers' education, wealth and knowledge in the field of the environment, etc. Similar to this result, compared to the consumers who are on a medium level salary scale, the consumers who are on a higher salary scale were more sensitive

to the convenience of recycling facilities in the process of reusing mobile phones. The best findings of the study show that consumer's attitude towards reusable mobile phones is completely willing to participate. The study finds out that those who used to live in the metropolitan area have deeper perceptions and attitudes about trading reusable mobile phones.

In addition, environmental protection decisions had a direct, positive effect on the trading of reusable mobile phones, which was consistent with others' studies [28,29]. Furthermore, public literacy also indirectly affected support for recycling facilities and services mediated by perceived impacts indirectly. This implied that consumers who are more devoted to protecting the environment and have a public reality tended to think that trading reusable mobile phones would bring more benefits and that they support recycling development more.

### 5.2. Practical Implications

Based on the above analysis and discussion of the research results, we have put forward some suggestions regarding urban e-waste trading. From a managerial perspective, these findings also have practical implications for reusable mobile phones, recycler planning and management policy.

It is suggested that to encourage consumers' support for trading reusable mobile phones, trading plans should aim to consider consumers' awareness and use. It is necessary to strengthen consumer awareness campaigns on the trading of reusable mobile phones through various channels. Relevant recycling companies can make full use of TV and other media, billboards, mobile clients and mobile Internet to spread the knowledge of trading of reusable mobile phones and encourage a green and civilized lifestyle. Meanwhile, mobile phone recycling companies will produce promotional advertisements on a large scale, which will affect the use of mobile phones for minors. Parents should try to avoid the excessive use of mobile phones by minors who have no self-control.

Compared with other countries' laws and regulations on reusable mobile phones, China has not established sufficiently effective laws and mandatory restraint policies for electronic waste recycling management. For example, the Solid Waste Pollution Prevention Law and the Circular Economy Promotion Law do not impose restrictions on the laws and regulations of consumers who are producers of reusable mobile phones, resulting in little voluntary trading. Therefore, the government needs to take measures to strengthen the top-level design of urban reusable mobile phone recycling and formulate effective laws and regulations. Public impact will affect consumers' initial willingness, and suggest that enterprises use social influence to guide consumer behavior decision-making to establish a virtuous circle of mobile payment community ecosystem. At the same time, it is recommended that the state introduce special e-waste management, further clarify the supervision system, consumer responsibility and obligation system, reward and punishment support mechanism, and credit system.

### 5.3. Limitations and Future Research

The empirical study has several limitations. First, the findings are not generalizable to all urban agglomerations. The sample is derived from a mega city with an orientation toward development. However, almost all results correspond to earlier findings in the literature, suggesting the modeled interrelationships may exist across the populations of consumers. Second, the latent factor participation is informed by binary variables as opposed to continuous or ordinal variables with a greater capacity to indicate variability. According to the confirmatory factor analysis, the factor is characterized by the lowest composite reliability coefficient and two of the weakest path loadings in the measurement model. Its relatively poor internal consistency is perhaps also attributable to the specific character of the survey questions (e.g., 'Reusable mobile phone trading saves resources.'). A superior conceptualization of the factor may allow further improvements in model fitness. Third, the structural equation model is estimated with cross-sectional data which neglects the dynamic relationship of public literacy and consumer trading returns. The static nature of the model implies that caution is warranted when interpreting the estimated relationships. In future research, the linear or non-linear relationship between various elements will be considered. Fourth, the observed factor of encouragement is informed by one

Likert statement ('A paid treatment will encourage us to adopt it'). While the answer provides a static interpretation, encouragement is a dynamic concept which is ideally measured with time-series data.

From a future development perspective, among the factors affecting the trading behavior of consumers' reusable mobile phones, environmental awareness, information leakage sensitivity, trading convenience, and consumer trading returns are important in the processing of external factors. There is an urgent need for the government and recyclers to publicize and educate consumers and the public about the popularization of new electronic products, such as mobile phones, through publicity media, mobile clients, and advertising. Secondly, the trading demand of reusable mobile phones is generally biased towards information security, channel convenience and high value. The consumers' trading expectation and recycling status cannot reach a unified consensus; the transparency of trading channels is the consumers' sense of security. With the popularization of environmental protection knowledge of mobile phones and the increase in environmental awareness, the awareness of young people with low education background and elderly people who are not engaged in environmental protection of trading reusable mobile phones will also be greatly improved. Finally, the degree of matching of consumer trading returns has an important consolidating effect on consumer trading perceptions. Extensively publicity and education activities on the recycling of reusable mobile phones should be carried out by various channels. Future urban comprehensive management laws and regulations should be improved.

**Author Contributions:** Conceptualization, Y.G. (Yifan Gu) and Y.W.; methodology, Y.W.; software, Y.G. (Yifan Gu); validation, X.Z. and Y.G. (Yu Gong); formal analysis, Y.W.; investigation, Q.Y. and Y.G. (Yifan Gu); resources, Y.W.; data curation, Y.G. (Yifan Gu) and Q.Y.; writing—original draft preparation, Q.Y.; writing—review and editing, Y.W. and Y.G. (Yifan Gu); supervision, Y.G. (Yifan Gu) and Y.W.; project administration, Y.G. (Yu Gong) and Y.W.; funding acquisition, Y.W. All authors have read and agreed to the published version of the manuscript.

**Funding:** The authors gratefully acknowledge the financial support from the National Key Technology R&D Program of China (No. 2018YFC1903106, 2018YFC1903603), National Natural Science Foundation of China (No. 41901240), Science and technology program of Beijing Municipal Education Commission (KM201910005008).

**Conflicts of Interest:** The authors declare no conflict of interest.

## Appendix A

QUESTIONNAIRE
INFORMATION
What is your gender?
a. Male; b. Female
What is your monthly household income level?
a. Under 5000; b. 5000–8000; c. 8001–20,000; d. 20,000 and above.
What is your age?
a. Under 18; b. 18–30; c. 31–60; d. 61 and above.
What is your education level?
a. Junior high school and below; b. High school, secondary school; c. University specialties, undergraduate; d. Graduate and above.
What is your professional status?
a. Environmental protection industry; b. IT; c. Business or service industry; d. Financial industry; e. Student; f. Other.
What is your number of phones in use?
a. 0; b. 1; c. 2; d. 3 and above.
BEHAVIOR ALATTITUDE
*Active Willingness to Vote* (7-point Likert scale used) (1 = completely disagree; 7 = completely agree)

- Adopting reusable mobile phones will help us enhance the quality of the environment at home.
- Adopting reusable mobile phones at home will contribute to environmental protection.

*Environmental Protection* (7-point Likert scale used) (1 = completely disagree; 7 = completely agree)

- For saving resources to contribute to environmental protection.
- For individual to contribute to reducing environmental hazard.

  *Public Literacy* (7-point Likert scale used) (1 = completely disagree; 7 = completely agree)

- I will reduce the environmental hazard of discharging untreated reusable mobile phones into the environment by adopting trading.
- Reusable mobile phones trading saves resources.

  *Consumer Trading Returns* (7-point Likert scale used) (1 = completely disagree; 7 = completely agree)

- A paid recycle treatment will encourage us to adopt it.
- I sell reusable things in order to obtain economic benefits.

SUBJECTIVE NORM

  *Environmental Policy Constraint* (7-point Likert scale used) (1 = completely untrue; 7 = completely true)

- I have the support of the family to reuse the reusable mobile phones.
- Reusable mobile phone recycling laws and regulations can play a constraining role for me.
- I am satisfied to participate in reusable mobile phone trading.

  *Neighbors Behavior* (7-point Likert scale used) (1 = completely disagree; 7 = completely agree)

- What my family think we should do is important to me.
- What my neighbors think we should do is important to us.

  *Family Member Influence* (7-point Likert scale used) (1 = completely untrue; 7 = completely true)

- I have the support of the family to adopt reusable mobile phones.
- Our relatives support the idea of us adopting the reusable mobile phones.

  *Promote Environmental Education* (7-point Likert scale used) (1 = completely disagree; 7 = completely agree)

- I feel morally obliged to reduce the volume of untreated mobile phones discharged into the environment.
- I feel guilty if we discharge untreated mobile phones into the environment.

PERCEPTUAL BEHAVIOURAL CONTROL

  *Specification of Recycling Channel* (7-point Likert scale used) (1 = completely disagree; 7 = completely agree)

- I will adopt a reusable treatment if it will be used at a low cost.
- I will trade reusable mobile phones actively if we are comfortable with the recycle technology.

  *Trading Determination* (7-point Likert scale used) (1 = completely disagree; 7 = completely agree)

- Our family will feel better if we contribute to prevent environmental pollution.
- Collecting reusable mobile phones takes up a lot of storage space in my house.

  *Active Trading Behavior* (7-point Likert scale used) (1 = completely disagree; 7 = completely agree)

- A simple and easy to operate procedure will encourage us to recycle.
- I produce less reusable mobile phones, no need for separation and trading.

RECYCLING FACILITIES AND SERVICES

  *Recycling Facility Convenience* (7-point Likert scale used) (1 = completely disagree; 7 = completely agree)

- There are trading bins in the community, with clear identification and close distances.
- A free mobile phone treatment will encourage us to trade it.

*Trading Convenience* (7-point Likert scale used) (1 = completely disagree; 7 = completely agree)

- An Internet technology will facilitate our selection of a recycle and reuse treatment.

*Recovery Time Cost* (7-point Likert scale used) (1 = completely disagree; 7 = completely agree)

- A low-time-cost recycle treatment will encourage us to adopt it.

*Information Leak Sensitivity* (7-point Likert scale used) (1 = completely disagree; 7 = completely agree)

- I attach great importance to the disclosure of information during the transaction.

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
