# Peer review of "Analysis of the Influence Mechanism of Consumers’ Trading Behavior on Reusable Mobile Phones"

_sustainability, doi:10.3390/su12093921_

Round 1
Reviewer 1 Report
Dear authors
The problem is very interesting, and the authors was very clear about that. But the main flaws on this paper are on the presentation of the results and also on the conclusions.
These flaws are considered to be huge, in order to accept your paper.
For instance, can the authors explain this age classes created (18-30; 30-60), because they aren’t muthual exclusive. For instance if I have 30 years I can be in both classes. Please explain.
On Table 4. Model fit test results, how can the authors state that the model has a “hight data reliability” with that fit values (see the book from Joseph F Hair, Barry J. Babin, Rolph E. Anderson, William C. Black.(2018) - Multivariate Data Analysis, 8th Edition. Print ISBN: 9781473756540)
Also the authors should perform two more tests: average variance extracted (AVE) and composite reliability(CR). They are not presented on this paper, and they are the most robust tests for SEM.
The conclusions are very poor, it should be more developed. The authors can easily related and discussed the conclusions with the theory background presented.
Author Response
Dear Reviewer #1:
Thank you for your letter and for the reviewers’ comments concerning our manuscript entitled Analysis of the Influence Mechanism of Consumers' Trading Behavior on the Recycling of Reusable Mobile Phones (ID: sustainability-766962). Those comments are all valuable and very helpful for revising and improving our paper, as well as an important guide significant to our research. We have studied the comments carefully and have made corrections which we hope will meet your approval. Revised portions are marked in red in the paper. The main corrections in the paper and the response to the reviewer’s comments are as attached.

Reviewer 2 Report
It is an interesting paper that uses a clear and well exposed methodology. The bibliography is extensive and with many recent articles. In general, there is nothing that can be considered incorrect. Only two small suggestions for improvement are made:
- At the end of the Introduction section, indicate the sections that the paper has, to anticipate the structure of the article.
- The research questions are not well developed. It would be necessary to clarify the objectives and argued. A case study is developed (but is so local) and it is focused on the peculiarities on the one specific country, China. It would be interesting analyse if cultural differences can affect the results.
- At the end of the discussion and conclusion section it would be interesting deep in managerial implications, limitations and future research lines.
Author Response
Response to Reviewer 2 Comments
Point 1: At the end of the Introduction section, indicate the sections that the paper has, to anticipate the structure of the article.

Response 1:
Added: Line 57-62
Thank you very much for your suggestion. We have made corresponding changes that added new part of the structure of the article at the end of the Introduction section. Please find the attached revised version.
Point 2: The research questions are not well developed. It would be necessary to clarify the objectives and argued. A case study is developed (but is so local) and it is focused on the peculiarities on the one specific country, China. It would be interesting analyse if cultural differences can affect the results.
Response 2:
Added: Line 57-62
Modified: Line 46-51, Line 303-305
Thank you very much for giving us a chance to revise our manuscript. We quite appreciate the reviewer’s careful evaluations and the insightful comments. According to the reviewer’s suggestion, we have revised the introduction and discussion and added more related literature, emphasized the objectives and argued at introduction and discussion. China is a country with many elements, all characteristics of it cannot be fully covered in this study. So, we will analyse cities included overseas and domestic with different characteristics in future research.
Point 3: At the end of the discussion and conclusion section it would be interesting deep in managerial implications, limitations and future research lines.
Response 3:
Added: Line 309-323, Line 333-369
Modified: Line 301-308, Line 324-332
Thanks for the good suggestion by the reviewer. According to the reviewer’s suggestion, we added and modified the conclusion of this study. We have split the conclusion into four sub-sections: discussion of results, practical implications, limitations and future research. Please find the attached revised version.
Reviewer 3 Report
Dear author/s,
the paper is dealing with an up-to-date topic and is relevant for the journal´s scope.
I think the paper can be of good quality, but a sufficient description of some of your methods is missing. For this reason, I have the following remarks:
a. You research mainly "consumer´s reusable mobile trading behavior", which means that recycling is only one of the research areas investigated. Therefore, the main goal (as well as title and everywhere it is appropriate) should be defined rather this way (inclusion of recycling makes it complicated to understand).
b. The questionnaire should be attached. Otherwise, it is very hard to assess the quality of research.
c. The variables described in Tab 1 are each from different literature sources. This, on one hand, show that your research review is sound, however, on the other side, it can cause a problem with the consistency of your analysis. Nevertheless, I cannot assess it, because the questionnaire is not presented.
d. In the part of the discussion, please provide more comparison with other studies. In the part of the conclusion, please provide some future research directions.
Author Response
Response to Reviewer 3 Comments
Point 1: You research mainly "consumer´s reusable mobile trading behaviour", which means that recycling is only one of the research areas investigated. Therefore, the main goal (as well as title and everywhere it is appropriate) should be defined rather this way (inclusion of recycling makes it complicated to understand).
Response 1:
Modified: Full Manuscript
Great thanks for the reviewer’s careful comment. "recycling" is a modification of corporate government implementation measures, "trading" is aimed at consumers, because the description of the two words in this article is not accurate enough, which affects the commenters' understanding of the purpose of this article We are sorry for these mistakes, and we made corresponding revisions in the title and the main text. Please find the attached revised version.
Point 2: The questionnaire should be attached. Otherwise, it is very hard to assess the quality of research; The variables described in Tab 1 are each from different literature sources. This, on one hand, show that your research review is sound, however, on the other side, it can cause a problem with the consistency of your analysis. Nevertheless, I cannot assess it, because the questionnaire is not presented.
Response 2:
Added: Line 381, Line 347-362
Thanks for the good suggestion by the reviewer. According to the reviewer’s suggestion, we have added above mentioned relevant questionnaire in the post-text and the ‘Appendix A’ of our attached revised version. In the conclusion of the article, discussing possible related issues with the consistency of your analysis.
Point 3: In the part of the discussion, please provide more comparison with other studies. In the part of the conclusion, please provide some future research directions.
Response 3:
Added: Line 309-323, Line 333-369
Modified: Line 301-308, Line 324-332
Thanks for the good suggestion by the reviewer. According to the reviewer’s suggestion, we added and modified the conclusion of this study. We have split the conclusion into four sub-sections: discussion of results, practical implications, limitations and future research. Please find the attached revised version.
QUESTIONNAIRE
INFORMATION
What is your gender?
a.Male; b.Female
What is your monthly household income level?
a.0-5000; b.6000-8000; c.9000-20000; d.30000 and above
What is your age?
a.Under 18; b.18-30; c.31-60; d.61 and above
What is your education?
a.Junior high school and below; b.High school, secondary school; c.University specialties, undergraduate; d.Graduate and above
What is your professional status?
a.Environmental protection Industry; b. IT; c. Business or service industry; d. Financial industry; e. Student; f. other
What is your umber of phones in use?
a.0; b.1; c.2; d.3 and above
BEHAVIOR ALATTITUDE
Active Willingness to Vote (7-point Likert scale used) (1 = completely disagree; 7 = completely agree)
- Adopting reusable mobile phones will help us enhance the quality of environment at home
- Adopting reusable mobile phones at home will contribute to environmental protection
Environmental Protection (7-point Likert scale used) (1 = completely disagree; 7 = completely agree)
- For saving resources to contribute to environmental protection is
- For individual to contribute to reducing environmental hazard is
Public Literacy (7-point Likert scale used) (1 = completely disagree; 7 = completely agree)
- I will reduce its environmental hazard of discharging untreated reusable mobile phones into the environment by adopting trading
- Reusable mobile phones trading to saving resources is
Consumer Trading Returns (7-point Likert scale used) (1 = completely disagree; 7 = completely agree)
- A paid recycle treatment will encourage us to adopt it
- I sell reusable things in order to obtain economic benefits.
SUBJECTIVE NORM
Environmental Policy Constraint (7-point Likert scale used) (1 = completely untrue; 7 = completely true)
- I have the support of the family to reuse the reusable mobile phones
- Reusable mobile phones recycling laws and regulations can play a constraining role for me.
- I'm satisfied to participate in reusable mobile phones trading.
Neighbours’ Behaviour (7-point Likert scale used) (1 = completely disagree; 7 = completely agree)
- What my family think we should do is important to me
- What my neighbours think we should do is important to us
Family Member Influence (7-point Likert scale used) (1 = completely untrue; 7 = completely true)
- I have the support of the family to adopt the reusable mobile phones
- Our relative support the idea of us adopting the reusable mobile phones
Promote Environmental Education (7-point Likert scale used) (1 = completely disagree; 7 = completely agree)
- I feel morally obliged to reduce the volume of untreated mobile phones discharged into the environment
- I feel guilty if we discharge untreated mobile phones into the environment
PERCEPTUAL BEHAVIOURAL CONTROL
Specification of Recycling Channel (7-point Likert scale used) (1 = completely disagree; 7 = completely agree)
- I will adopt a reusable treatment if it will be used at few cost
- I will trade reusable mobile phones actively if we are comfortable with the recycle technology
Trading Determination (7-point Likert scale used) (1 = completely disagree; 7 = completely agree)
- Our family will feel better if we contribute to preventing environmental pollution.
- Collecting reusable mobile phones takes up a lot of storage space in my house.
Active Trading Behaviour (7-point Likert scale used) (1 = completely disagree; 7 = completely agree)
- A simple and easy to operate procedure will encourage us to recycle
- I produce less reusable mobile phones, no need for separate and trading.
RECYCLING FACILITIES AND SERVICES
Recycling Facility Convenience (7-point Likert scale used) (1 = completely disagree; 7 = completely agree)
- There are trading bins in the community, with clear identification and close distances.
- A free mobile phones treatment will encourage us to trade it
Trading Convenience (7-point Likert scale used) (1 = completely disagree; 7 = completely agree)
- An Internet technology will facilitate our selection of a recycle and reuse treatment
Recovery Time Cost (7-point Likert scale used) (1 = completely disagree; 7 = completely agree)
- A few-time-cost recycle treatment will encourage us to adopt it
Information Leak Sensitivity (7-point Likert scale used) (1 = completely disagree; 7 = completely agree)
- I attach great importance to the disclosure of information during the transaction

Round 2
Reviewer 3 Report
Thank you for the revisions made, I believe it increased the quality of your paper.
However, some minor revisions are still needed. For example, you corrected the salary range, but some values are missing there right now (from 5,000 to 6,000 etc.); the numbers of proportion (Table 2) are some presented with two decimal numbers and some with one (should be unified).
I recommend carefully review the understanding, stylistics, and content once again.
Author Response
Response to Reviewer 3 Comments
Point 1: However, some minor revisions are still needed. For example, you corrected the salary range, but some values are missing there right now (from 5,000 to 6,000 etc.); the numbers of proportion (Table 2) are some presented with two decimal numbers and some with one (should be unified). I recommend carefully review the understanding, stylistics, and content once again.
Response 1:
Modified: Line:195,200,210,236,254,259,296,298
Great thanks for the reviewer’s careful comment. We quite appreciate the reviewer’s careful evaluations and the insightful comments. According to the reviewer’s suggestion, we have revised the table in the manuscript and the numbers of proportion in the text are presented with three decimal numbers.